# Fatal Hypernatremic Dehydration in a Term Exclusively Breastfed Newborn

**DOI:** 10.3390/children9091379

**Published:** 2022-09-13

**Authors:** Christie del Castillo-Hegyi, Jennifer Achilles, B. Jody Segrave-Daly, Lynnette Hafken

**Affiliations:** 1Department of Emergency Medicine, CHI St. Vincent, Little Rock, AR 72205, USA; 2Fed is Best Foundation, Little Rock, AR 72223, USA; 3TelePeds, Santa Fe, NM 87505, USA; 4Holy Cross Hospital, Silver Spring, MD 20910, USA

**Keywords:** neonatal hypernatremic dehydration, neonatal dehydration, exclusive breastfeeding, neonatal brain injury, developmental disability, Baby-Friendly Hospital Initiative, epilepsy, acute kidney injury, intracranial hemorrhage

## Abstract

Hypernatremic dehydration in term newborns has steadily increased in incidence with increasing efforts to promote exclusive breastfeeding before hospital discharge, a key metric of the Baby-Friendly Hospital Initiative. The following report details a case of a term newborn infant who had evidence of poor intake while exclusively breastfeeding during his hospital stay that may not have been recognized by health care providers. The infant was discharged home and was subsequently found by the parents in cardiac arrest 12 h after discharge and was found to have hypernatremic dehydration. Although return of spontaneous circulation was achieved after fluid resuscitation, the infant sustained extensive hypoxic-ischemic brain injury due to cardiovascular collapse. Due to the infant’s extremely poor prognosis, life support was withdrawn at 19 days of age and the infant expired. This sentinel case demonstrates multiple pitfalls of current perceptions of normal vs. abnormal newborn feeding behavior, weight loss percentages, elimination patterns, and acceptable clinical thresholds believed to be safe for neonates. Newer data have shown that hypernatremia occurs commonly in healthy, term breastfed newborns at weight loss percentages previously deemed normal by most health professionals and hospital protocols. In-hospital strategies to prevent excessive weight loss and screening for hypernatremia in response to signs of inadequate feeding have the potential to prevent tens of thousands of readmissions for feeding complications a year, as well as hundreds of millions in health care costs.

## 1. Introduction

Neonatal hypernatremic dehydration (NHD) is a brain– and vital organ–threatening, potentially lethal condition in neonates defined as a sodium of >145 mEq/L [1] Currently, it is most commonly associated with inadequate feeding, especially in exclusively breastfed infants in the first days to weeks after delivery. Serum sodium levels between 145 and 149 mEq/L are considered mild hypernatremia; 150–160 mEq/L moderate hypernatremia; and above 160 mEq/L severe hypernatremia [1]. Increased risk of early developmental delay and radiographic evidence of injury have been detected in infants developing moderate hypernatremia greater than 150 mEq/L [2,3]. Moderate to severe hypernatremia can lead to brain hemorrhage, brain edema, cerebral infarction, acute kidney injury, hepatic injury, peripheral venous and arterial thrombosis, disseminated intravascular coagulation, seizures, and death [3,4]. Long-term sequelae of NHD include moderate to severe neurodevelopmental delay [5]. 

Exclusive breastfeeding from birth to 6 months is recommended by the WHO, UNICEF, CDC, and the AAP due to studies showing an association between breastfeeding and improved infant health outcomes, such as reduced rates of gastrointestinal infections, upper and lower respiratory infections, otitis media, asthma, eczema, leukemia, obesity, SIDS, and infant mortality [6]. Step Six of the Ten Steps to Successful Breastfeeding, which recommends to “give no food or fluid other than breast milk unless medically indicated,” is a key hospital metric of the Baby-Friendly Hospital Initiative [7]. However, while the majority of healthy, term infants can be exclusively breastfed without need for supplementation, it is important to identify the infants who are at risk for serious complications such as hypernatremic dehydration if not supplemented in a timely and adequate manner. More recent data suggest that more infants may require medically indicated supplementation for conditions such as hypernatremia than previously estimated.

The following is one such case of an infant exhibiting multiple signs of poor feeding and likely evolving hypernatremic dehydration during their birth hospitalization. Information was obtained through careful examination of the medical record and autopsy report, and through direct correspondence with the mother. This case ultimately resulted in cardiovascular collapse at home shortly after discharge, which likely resulted in extensive hypoxic-ischemic brain injury not compatible with long-term survival, eventually leading to withdrawal of life support and death. Careful attention is given to the characteristics of the breastfeeding sessions before discharge as well as the behavior of the infant during their hospitalization and upon discharge. The aim of this work is to review the current literature on hypernatremic dehydration, to examine the events leading up to the development of cardiovascular collapse from hypernatremia, and to offer potential strategies to prevent infant harm from hypernatremia.

## 2. Case Report

The patient was a term male neonate born at 39 weeks 2 days via cesarean delivery due to fetal intolerance of labor as evidenced by prolonged fetal heart rate decelerations. The mother was a 31-year-old primiparous female, gravida 2, para 0 with a history of one spontaneous miscarriage and several risk factors for low milk supply and delayed lactogenesis II, including primiparity, prior lap band surgery, BMI of 43, diet-controlled gestational diabetes, PCOS, and history of fertility problems. After delivery, the infant cried immediately and was given Apgar scores of 8 and 9. He developed transient tachypnea that resolved after 10 min of CPAP. The neonate was given a 35 mL normal saline bolus for presumed hypovolemia and acidosis related to fetal intolerance of labor. The infant weighed 3360 g and he was transferred to the special care nursery (SCN) after delivery, where hourly point-of-care glucose monitoring found glucose levels of 109, 107, 103, and 85 mg/dL. Glucose monitoring ended once the neonate was transferred from the SCN after a few hours to the mother–baby unit. 

The mother had attended the hospital breastfeeding classes at her Baby-Friendly certified hospital and her intention was to exclusively breastfeed from birth. Given the need for infant stabilization, her first breastfeeding attempt occurred 2.5 h after birth, which the infant tolerated given stable glucose levels through the first direct breastfeeding session. The first session lasted 40 min, was observed by the SCN nurse, and was given a LATCH score of 6 of out 10 [8]. A LATCH score of 0–3 is regarded as poor, 4–7 as moderate, and 8–10 as good [9]. There were repeated attempts to hold the nipple in the mouth to stimulate suck but eventually the mother succeeded and required minimal assistance. 

There were 14 breastfeeding sessions in the first 24 h lasting 12–40 min each, mostly occurring between 25 min to 2 h apart (two outlier intervals of 3 and 6 h). At 7 h of life, the infant was weighed and had lost only 1.2% of birth weight. The LATCH score was rated a 10 by the IBCLC the next morning at almost 8 h after birth. In addition to the multiple risk factors for breastfeeding challenges listed above, the IBCLC noted risk factors for insufficient glandular tissue from the breast exam including small, widely spaced breasts that experienced no change during pregnancy. The nipples were slightly everted and expressible colostrum was observed. It was at this time that the mother “expressed concern about [her] baby feeding for a long time”. The breastfeeding latch was thus directly observed for 35 min and was described as “non-nutritive sucking”. However, the infant’s rooting and sucking were noted to be within normal limits and the baby appeared sleepy and “satisfied” after the feed. The patient was reassured that breastfeeding was going well. 

At around 24 h, there were 3 recorded sessions of poor feeding where repeated attempts to maintain the nipple in the mouth and stimulate sucking failed, and the LATCH score was rated a 7. Two more feeding sessions were attempted 45 and 20 min after that, only lasting 2 and 10 min, respectively. Shortly after this, the feeding durations increased significantly with most sessions lasting 60 min and as high as 90 min. According to the mother, her infant was fussy and crying unless he was latched at the breast. Several feeding sessions were not recorded due to near-continuous feeding in response to persistent hunger cues. At 27 h of life, the infant was found to be 4.76% below birth weight with 3 wet diapers and 6 soiled diapers in the previous 24 h.

The IBCLC evaluation on the second day after birth noted that she was “unable to express breast milk”. It was at this time that the mother was informed that her PCOS may put her at risk of problems with milk supply and was told she may benefit from herbal galactagogues. According to the mother, she was encouraged to solely breastfeed during her hospital stay and that there should be no supplementation with formula.

Between 51 h to the time of discharge at around 63 h, the infant was noted multiple times to be “cluster feeding,” breastfeeding for 60 min or greater, or near-continuously for several hours with an excellent LATCH score of 10 through the time of discharge. At 53 h of life, the infant was found to have 9.72% of birth weight loss (>75 percentile on the Newborn Weight Loss Tool), 2 wet diapers and 2 soiled diapers, and normal transcutaneous bilirubin of 2.6 mg/dL. 

The last evaluation by an IBCLC at around 63 h, shortly before discharge, noted the “baby feeding frequently and very fussy when not on breast,” and that the baby slept for only 3 h. Baby was noted to have 9.9% weight loss although it was unclear when this was recorded. The record showed 4 wet diapers and 11 soiled diapers in the second 24 h. She was scheduled for a follow-up appointment the next morning. According to mother, she was not encouraged to start supplemental feedings upon discharge that afternoon at around 64 h of life. She expected the weight check to occur the next day at her pediatrician follow-up to determine the need for supplementation. The baby was still latched at the breast while the mother was being transported to the hospital entrance upon discharge.

At home, the infant continued to breastfeed near continuously and cried whenever he was not latched at the breast. According to the mother, this continued until 2 am when he eventually fell asleep. She states she had put the baby down in his crib and sat in her bed to read her breastfeeding information pamphlet. She then unintentionally fell asleep while reading and woke up 30–45 min later, at 76.5 h of life, to find him cyanotic, limp, and unresponsive, and called 911. Conflicting reports in the medical record and autopsy suggested the possibility of unintentional co-sleeping while breastfeeding immediately prior to finding the infant unresponsive, which is a known risk during breastfeeding and skin-to-skin care in the immediate postnatal period made worse by maternal exhaustion [10]. However, according to the mother, she found her infant supine in his crib, with no overlying blankets or objects. The father, a trained emergency medical technician, initiated chest compressions immediately. 

EMS transported the infant while continuing chest compressions and bag-valve ventilations. Prior to arrival, EMS placed an intraosseous line and started infusing normal saline. His initial rhythm was asystole. Upon arrival to the emergency department, the record showed pulseless electrical activity. The infant was intubated. The infant received 7 weight-appropriate doses of intravenous epinephrine, continued normal saline infusion (55 mL in the ER, unknown amount in the field), and full resuscitation. The infant was hypothermic with a temperature of 93.1 °F. After 30 min of resuscitative effort, cardiac activity was checked by echocardiogram and no cardiac activity was found. The family was informed of his moribund condition and with parental consent, chest compressions were terminated while they were allowed to be with their infant. During this time, although chest compressions were stopped, mechanical ventilation and IV fluids were continued. Twenty minutes later, it was discovered that the infant had regained his pulse, was in sinus rhythm with a heart rate of 120 and improvement in skin color. This occurred approximately 1.5 h after the discovery of his cardiac arrest at home. He received a 5 mL 50% dextrose bolus after the return of circulation, but no glucose was checked. The infant was subsequently transferred to the nearest Level III NICU.

It is unclear what his weight was around the time of cardiac arrest due to the emergent need for IV fluids before ED arrival and during his ED resuscitation. It is also unclear what his blood glucose was upon presentation given no glucose level was recorded until 4–5 h after he received a 5 mL bolus of 50% dextrose and a 10% dextrose infusion had been started. His first glucose measurement at that time was 110 mg/dL. The initial laboratory data 6 h after the start of resuscitative treatment revealed an elevated sodium of 155 mEq/L, potassium 3.3, chloride 118, HCO_3_ 13, lactate 6.4. The ammonia level was normal. The infant had a WBC of 27.1, Hgb 14, Hct 41, platelets 340. Serum bilirubin was within normal limits. His infectious work-up including blood cultures, CSF cultures, and CSF HSV PCR were all negative. The infant’s newborn screening panels, including galactosemia, acylcarnitine and amino acid panels, were normal. The infant was treated with a hypothermia cap for 72 h upon admission.

The infant’s post resuscitation course was significant for gradual correction of hypernatremia followed by hyponatremia presumed to be caused by Syndrome of Inappropriate Anti-Diuretic Hormone (SIADH). Neurological consultation was obtained for presumed post-arrest hypoxic-ischemic encephalopathy and seizure-like activity. The infant exhibited repeated diaphragmatic contractions, continuous rhythmic bilateral ankle shaking, and a 30-min episode of bilateral hand shaking. The 3-day EEG showed baseline continuous suppression without any background features consistent with severe encephalopathy. Very frequent electrographic seizures were present (2 to 7 per hour), lasting 2 to 5 min, which were treated with phenobarbital. The brain MRI showed diffuse signal abnormalities in the basal ganglia and bilateral thalami, consistent with significant hypoxic-ischemic injury to the deep gray nuclei and portions of the brainstem. 

His physicians discussed the infant’s grim neurological prognosis with the parents, and they agreed to proceed with extubation and withdrawal of life support at 19 days of life, after which the infant expired. An autopsy was performed, and the causes of death were determined to be hypernatremic dehydration, resulting in cardiac arrest, causing hypoxic-ischemic encephalopathy and concurrent pneumonia, the latter likely the result of prolonged mechanical ventilation. According to the autopsy, asphyxiation around the time of the initial cardiac arrest could not be proven or disproven. However, the lack of response to ventilatory support and the delayed recovery of spontaneous circulation in response to fluid resuscitation strongly suggests hypotensive hypernatremic dehydration as a significant contributor to cardiovascular collapse. 

## 3. Discussion

### 3.1. Mechanism of Disease

Serum sodium concentration is usually tightly regulated to maintain cellular integrity between 135–145 mEq/L. High sodium concentrations increase serum osmolality resulting in cell shrinkage followed by cell lysis if normal osmolality is rapidly restored during rehydration. Cytotoxic injury can result in direct brain and vital organ injury as well as vascular damage, thrombosis, infarction, and hemorrhage. In a study of hypernatremic neonates with a range 150–194 mEq/L, sequelae included brain hemorrhage, brain edema, cerebral infarction, acute kidney injury, liver failure, peripheral venous and arterial thrombosis, disseminated intravascular coagulation, and death [3,11]. Mortality rates increase with increasing severity of hypernatremia (see Table 1) [12]. Hypernatremia is commonly accompanied by excessive weight loss, hyperbilirubinemia, and hypoglycemia [13,14]. These co-morbid conditions compound the brain and vital organ injury caused by starvation and dehydration.

The main cause of breastfeeding-related hypernatremia is poor milk intake, which can be caused by low maternal milk supply, ineffective transfer of milk, or a combination of both, in the setting of ardent attempts at maintaining exclusive breastfeeding status. Poor milk supply can either be due to low or scant colostrum production, delayed onset of lactogenesis II (DLII), and/or inadequate mature milk supply [15,16]. 

Colostrum, the first milk present after birth, is typically low in volume and low in caloric density, containing 54 kcal/dL, in contrast to mature breast milk, which typically contains 66 kcal/dL [17]. A quantitative study of colostrum production of healthy mothers delivering term infants showed an average total production of 56 mL in the first 24 h, 185 mL on the second day, 383 mL on the third, and 580 mL on the fourth day [18]. These average volumes provide a total of 30 kilocalories (kcal) on the first day, 100 kcal on the second day, 207 kcal on the third day, and exceeding a term neonate’s full 110 kcal/kg/day requirement on the fourth day, with 313 kcal [19]. Similarly, a neonate’s maintenance fluid requirement of 100 mL/kg/day is not met until the third day under average conditions. Therefore, postnatal exclusive colostrum feeding results in a “time-limited period of underfeeding” according to the Academy of Breastfeeding Medicine [20]. An exclusively breastfeeding neonate responds to this low fluid and caloric intake with weight loss, low urine output (1–2 wet diapers/day), compensatory fluid shifts, catabolism of internal caloric reserves, rising sodium, declining glucose levels, and ketosis [21]. This also commonly leads to hunger and frequent and longer durations of nursing in the first days referred to as “cluster feeding” or “Second Night Syndrome”. This process usually reverses once a mother transitions from producing colostrum to transitional milk to full production of mature milk, known as lactogenesis II. 

This physiology is corroborated by a study carried out by Futatani et al. showing that among healthy, term exclusively breastfed newborns without excessive weight loss (>10%), serum glucose gradually decreases, and serum sodium and beta-hydroxybutyrate (βOHB) increase until their magnitudes peak at 48–59 h of life, the period of maximum weight loss, reflective of the relative fasting conditions before the onset of lactogenesis II [21]. If this period of low intake is prolonged, as in the case of exclusive breastfeeding with delayed lactogenesis II, defined as occurring >72 h after delivery, it can exceed a neonate’s tolerance for weight loss and dehydration, resulting in hypernatremia and hypoglycemia. A study by the same group that included exclusively breastfed term infants with excessive weight loss of ≥10% found a strong positive correlation between βOHB levels and weight loss percentages (r = 0.740; *p* < 0.001), a strong positive correlation between βOHB and blood sodium levels (r = 0.703; *p* < 0.001), and a negative correlation between βOHB and glucose levels (r = −0.561; *p* < 0.001) indicating the co-existence of starvation and dehydration in hypernatremic neonates [22]. The study also found that sodium levels above 145 mEq/L and glucose levels < 47 mg/dL were common, affecting about a third of the studied cohort; 5% had sodium ≥ 150 mEq/L.

It is important to note that 72 h is an arbitrary definition, based on commonly observed patterns of lactation, and that the physiological definition of “delayed” is lactogenesis II that does not occur before the development of metabolic derangements associated with starvation and dehydration. Delayed onset of lactogenesis II in a mother strongly motivated to exclusively breastfeed can result in critical levels of hypernatremia during the period between hospital discharge and the first outpatient clinic visit, as in the infant discussed. If supplementation is discouraged upon discharge and lactogenesis II does not occur in time to prevent further weight loss, hypernatremia can progress to levels that can cause brain injury, cardiovascular collapse, and death. Failure to discuss the possibility of insufficient feeding due to delayed lactogenesis II in exclusively breastfed newborns, and failure to recommend supplemental milk in response to signs of inadequate feeding, may result in hypernatremia, as well as kernicterus and severe hypoglycemia [23,24]. This represents a major gap in patient education before discharge. 

### 3.2. Risk Factors 

Identified maternal risk factors for NHD include cesarean delivery, primiparity, breast anomalies or breastfeeding problems, excessive pre-pregnancy maternal weight, delayed first breastfeeding, and no prior breastfeeding experience, all of which were present in the case above (see Table 2) [25]. Primiparous mothers with no prior breastfeeding experience may be less likely to recognize signs of persistent infant hunger, especially if she is told by her health professionals that such signs are normal. There is significant overlap of these risk factors with those of delayed lactogenesis II, which include primiparity, cesarean section, longer duration of labor, maternal BMI > 27 kg/m^2^, flat or inverted nipples (which may lead to suboptimal latch), advanced maternal age ≥ 30 years, hypertension, gestational diabetes, and hypothyroidism, many of which are increasing in prevalence in the population of expectant mothers [25,26,27]. Delayed onset of lactogenesis II has been associated with higher maternal and infant stress during labor and delivery measured through cortisol levels in cord blood and maternal blood shortly after delivery [28].

Studies have shown that DLII and prolonged insufficient milk supply are common. DLII occurs in an estimated 22% of healthy mothers delivering healthy term babies. [15] In this study, neonates of mothers with DLII showed a 7-fold higher risk of excessive weight loss of >10%. Rates of DLII are even higher among primiparous mothers occurring in 42–44% [29,30]. A study of DLII in obese women (≥30 kg/m^2^) found an incidence of 57.9% [31].

Several reports of infants developing hazardous and fatal hypernatremia have occurred beyond the first week after birth due to exclusively breastfeeding with insufficient breast milk supply [32,33,34]. Quantitative studies on the prevalence of chronic milk insufficiency in the first month and beyond are limited; however, the available studies suggest insufficient milk supply is common, especially in the first month after delivery. One study showed that, in the first month after birth, an estimated 15% of healthy, primiparous mothers motivated to exclusively breastfeed had persistent low milk supply despite intensive lactation support and intervention [35]. A more recent quantitative study of maternal milk supply in the first month showed two-thirds of mothers having less than a full milk supply (defined as 440 mL/day for study purposes) in the first two weeks after birth and one-third having low milk supply in the two weeks following [36]. 

The infant in the case report is distinct from most reported cases of neonatal hypernatremia deaths, with more protracted courses presenting after the first week of life with >20% weight loss and sodium levels above 170. Those infants likely survived that degree of weight loss and hypernatremia by receiving some measure of mature breast milk once lactogenesis II occurred, which would provide higher volumes than colostrum, but lower volumes than what is required to maintain weight and hydration. Those infants likely experienced more gradual weight loss and rise in serum sodium to reach such extreme levels. The case mother never experienced lactogenesis II before the infant was found unresponsive. In addition, the medical record documented evidence of no expressible colostrum on the second day of life. Such an infant receiving little to no colostrum for 2–3 days, while expending energy to nurse continuously for more than 50 h before becoming lethargic, may have experienced near complete caloric and fluid deprivation, resulting in an accelerated course of weight loss, hypernatremia, and possibly hypoglycemia, leading to apnea and cardiovascular collapse.

### 3.3. Epidemiology

Hypernatremic dehydration, once considered a rare condition, has been rising in incidence since the 1990s, coinciding with global efforts to increase rates of exclusive breastfeeding before hospital discharge, a key hospital metric of the WHO/UNICEF Baby-Friendly Hospital Initiative, codified in U.S. hospitals by the Joint Commission Perinatal Core Measure-05 (PC-05) [37,38]. Observational data of one Jamaican hospital found a three-fold increase in NHD cases upon establishment of a BFHI program [39]. A U.K. study cited an incidence of hypernatremia defined as a serum sodium of ≥160 mEq/L of 1 in 1400 [40]. A U.S. study found an incidence of 1.9% among hospitalized term and near-term neonates [41]. Estimates of the true incidence of hypernatremic dehydration have been limited by inconsistent testing and lack of universally adopted clinical guidelines for hypernatremia screening. 

A 2018 study on healthy term newborns universally screened for hypernatremia in the first 72 h after birth confirmed disturbingly high rates of hypernatremia of >145 mEq/L in healthy, term infants, occurring in 30.9% [1]. In this mixed cohort of term neonates, 74.5% of hypernatremia cases occurred in exclusively breastfed neonates, 21.6% occurred in mix-fed neonates, and 3.9% occurred in formula-fed neonates. Within each feeding modality, hypernatremia occurred in 36.5% of exclusively breastfed neonates and 37.95% among mix-fed neonates, and in 6.25% among formula fed neonates, a 6-fold difference. Higher rates of NHD among mix fed infants may represent exclusively breastfed infants that developed medical indications for supplementation. The study authors concluded that the weight loss cut-off of 7–10% widely cited as the threshold for developing hypernatremia does not predict all cases of mild to moderate hypernatremia. The study identified several independent risk factors for hypernatremia and found that infants who were male, born by cesarean delivery to a multiparous mother with higher education, had the lowest weight loss cut-off for increased hypernatremia risk, occurring at 4.77% weight loss. Optimal weight loss thresholds that predicted higher risk for hypernatremia ranged from 4.77% to 10.8% depending on the number of risk factors. Other studies have shown similar results. A systematic review of 1485 cases of breastfeeding-related hypernatremia identified several cases of hypernatremia occurring below 5% weight loss, increasing in severity with increasing weight loss percent, 96% occurring at weight loss of ≥10% [25]. Another study showed 95% of hypernatremia cases occurring above 7% weight loss [2]. 

It is important to note that data from the Newborn Weight Loss study have shown that 10% of vaginally delivered and 25% of cesarean delivered term newborns lose greater than 10% of birth weight in the first 72 h of life [42]. Additionally, the study showed healthy, term vaginally delivered and cesarean delivered exclusively breastfed newborns had median weight loss percentages of 7% and 8%, respectively, a degree of weight loss already associated with increasing incidence of hypernatremia. This suggests that cases of hypernatremia in healthy, term exclusively breastfed newborns are likely much more common than previously estimated. This has important implications on how clinicians should interpret “normal” and “safe” weight loss for breastfed newborns given that safe weight loss should preclude the presence of serious metabolic abnormalities such as hypernatremia. The main limitation of the Newborn Weight Loss study is that no data on rates of hypernatremia in the cohort were available given that routine hypernatremia screening was not performed. Therefore, it cannot be assumed that this cohort of exclusively breastfeeding neonates used to determine “normal” distributions of weight loss over time were protected from clinically significant complications of inadequate milk intake. In fact, a follow-up study on the same population found that 4.3% of vaginally delivered exclusively breastfed newborns and 2.8% of cesarean delivered exclusively breastfed newborns were readmitted for complications such as jaundice and dehydration, which were both higher than those found in their formula-fed counterparts (2.1% and 1.5% respectively) [43]. Unsurprisingly, weight loss of >10% at 48–72 h was associated with a greater than 2-fold increased risk of readmission when compared to weight loss of <8%.

The infant in this case report’s last weight loss before discharge was 9.7% and due to the critical nature of his presentation, he was not weighed at first presentation in the emergency department before receiving an unknown amount of IV fluids. Before presentation, however, his weight loss pattern using the Newborn Weight Loss Tool followed a linear pattern of weight loss and if extrapolated to the time of cardiac arrest, it would have been around 14% weight loss. In our clinical experience, lethargy, seizures, and unstable vital signs are common around 15% weight loss caused by delayed lactogenesis II. While cases of fatal hypernatremia typically describe infants with >20% weight loss more than 7 days after birth, lethargy, seizures and apnea have been described in a 12-day and a 6-day-old infant with 15% and 18% weight loss, with sodium levels of 172 and 173 mEq/L, respectively [44]. Given the infant received an unknown amount of 0.9% saline 5 h before the first lab results, it is unclear how high the sodium level was at the time of arrest, possibly closer to 160–170 mEq/L, similar to other infants who developed similar weight loss, lethargy, and apnea. It is unclear if rapid correction of sodium levels contributed to the encephalopathy found on MRI. It is also unclear if hypernatremia was the sole cause of the infant’s cardiac arrest.

Alternatively, hypoglycemia may have been a contributing factor. Other infants have been described to develop lethargy from hypoglycemia due to poor breast milk intake and had similar weight loss percentages as the case infant. One case series describes 11 previously healthy, term, breastfed neonates without risk factors for hypoglycemia who developed symptomatic hypoglycemia below 36 mg/dL between days 2–5 [24]. The majority of infants presented lethargic and were poorly feeding. Five had respiratory compromise including shallow breathing, cyanosis, and apnea with glucose levels below 20 mg/dL. These infants had weight loss between 4.2% to 16% of their birth weight, similar to the case infant (one outlier had 0% weight loss, but only two had >10% weight loss). Five out of six brain MRIs obtained in the infants showed extensive injury to several lobes of the brain. Had the case infant not received adequate breast milk as evidenced by the hours of fussing, continuous breastfeeding for greater than 50 h starting the second day of life, no expressible colostrum on that second day, and hypernatremia, then both hypernatremia and hypoglycemia of less than 20 mg/dL may have contributed to hypotonia and prolonged apnea that could lead to hypoxic-ischemic encephalopathy and cardiac arrest. Other infants may experience similar outcomes due to inability to tolerate fasting conditions due to inborn errors of metabolism such as MCADD [45]. The case infant’s newborn screening tests were negative for all conditions including inborn errors of metabolism.

### 3.4. Signs and Symptoms 

Infants who develop hypernatremic dehydration will show signs of poor feeding including frequent, continuous and even high-pitched crying, frequent feeding (less than every 2 h), consistently prolonged feedings (30–60 min per feed), poor skin turgor, delayed capillary refill, jaundice, low to absent wet diapers, “red brick dust” or pink urate crystals in diapers, >5% weight loss, easy fatiguability during feeds, fever, and irritability [25,46]. Seizures, lethargy, inability to feed, sunken fontanelles and visible suture lines, apnea, bradycardia, and hypothermia are late signs that precede the onset of cardiovascular collapse (see Table 3) [47]. Reports of hypernatremia describe the common scenario of a highly educated mother who is motivated to comply with exclusive breastfeeding recommendations and is unaware of the signs of poor feeding [37].

Some infants may show few signs of active distress such as high-pitched crying, and “quietly starve,” losing weight while appearing sleepy and satisfied after breastfeeding, often not waking spontaneously to feed. These infants are at greater risk of developing hazardous hypernatremia and some have been found to be greater than 20% below their birth weight, hypotonic, hypothermic, and moribund [32,33,44,48]. Both parents and health professionals may fail to notice or adequately treat the slow progression of ongoing weight loss, starvation, and illness in an infant in the misguided effort to maintain exclusive breastfeeding status even when signs of suboptimal feeding are present. 

While diaper counts in exclusively breastfed newborns are commonly regarded as useful indicators of adequate versus inadequate feeding, a study on elimination patterns in exclusively breastfed newborns has shown that they are unreliable indicators of adequate feeding in the first 4 days of life [49]. Even neonates with weight loss of >10% can produce up to six wet and soiled diapers on day 4. Similarly, the infant in this case report produced wet and soiled diaper counts widely regarded as normal despite eventually developing hypernatremia. 

It is important to note that the 2009 Academy of Breastfeeding Medicine (ABM) guidelines for supplementation, which guided BFHI certified hospitals in the case described above stated “an infant who is fussy at night or constantly feeding for several hours” was not an indication for supplementation [50]. In addition, an infant developing 8–10% weight loss along with delayed lactogenesis II of >5 days of life is listed as a “possible indication for supplementation,” which the case infant had not yet surpassed. This may explain the reluctance of his health providers to offer supplementation despite hours of crying, continuous nursing, and 9.7% weight loss that the infant developed in the hospital. The newer 2017 ABM supplementation guidelines are similar, stating again that a fussy infant “constantly feeding for several hours” is not an indication for supplementation, with the additional caveat that “cluster feeding,” defined as “several short feeds close together,” is believed to be normal newborn behavior. The guidelines recommend a feeding evaluation to “observe the infant’s behavior at the breast…to ensure that the infant is latched deeply and effectively” [51]. Unfortunately, excellent latch, as seen in the described case, does not guarantee adequate breast milk intake. The newer supplementation guidelines also added >75th percentile weight loss on the Newborn Weight Loss Tool (NEWT) and “high sodium levels” as possible criteria for supplementation. However, it fails to offer a guideline of what clinical signs and percent weight loss warrants screening for hypernatremia. Given newer data on the high incidence of hypernatremia in healthy term newborns within the first 72 h at lower weight loss percentages than previously assumed, and the lack of hypernatremia data in the NEWT, these criteria may be inadequate to protect healthy term breastfed newborns from NHD and its short- and long-term sequelae.

Hyperbilirubinemia is a common co-morbid condition to hypernatremia. Interestingly, this case infant did not have hyperbilirubinemia. Not all hypernatremic neonates develop hyperbilirubinemia. A systematic review has shown that only 45% of hypernatremia cases have jaundice [25]. The literature has shown increased risk of hyperbilirubinemia among neonates of Asian, both Far East and Southeast Asian descent, and lower propensity among non-Hispanic White infants and African-American infants [52,53,54]. The case infant was non-Hispanic White and had minimal elevation of bilirubin. Therefore, it is important to not exclude the possibility of hypernatremic dehydration on the basis of no clinical jaundice or laboratory evidence of hyperbilirubinemia.

### 3.5. Long Term Sequelae

Survivors of NHD can develop moderate to severe long-term neurodevelopmental disabilities including intellectual impairment, other neurodevelopmental disabilities, and seizure disorders depending on the distribution of brain injury an affected infant sustains [5,55,56,57]. In a study of 100 consecutive neonates admitted for hypernatremic dehydration with sodium levels of ≥ 150 mEq/L, among 93 infants where brain MRI was obtained, an astonishing 45.2% had neuroradiological lesions, including brain edema, hemorrhage, and thrombosis [11]. One study has shown that among neonates who develop sodium levels ≥ 150 mEq/L, more than 50% had evidence of developmental delay on standardized developmental screening at 1 year of age [58]. Another prospective case-control study of 65 hypernatremic neonates (range 153–195 mg/dL) and 65 healthy controls followed to 24 months found that 25% of the NHD group had developmental delay on Denver II developmental assessment with corresponding values of 21% at 12 months, 19% at 18 months and 12% at 24 months of age, compared to 0.3% in the control group at similar ages [59]. Severity of hypernatremia was strongly correlated with poor developmental outcomes. Another smaller study on long-term outcomes found 47% of hypernatremic infants were affected with 5 out of 15 described as moderately disabled and 2 out of 15 as severely disabled [5]. The largest follow-up study to 36 months in 183 term infants with sodium ≥ 150 mEq/L found that 17.5% were diagnosed with abnormal developmental outcomes [56]. 

### 3.6. Prevention

Given an estimated 3.25 million term neonates born every year in the U.S., an estimated 35%, or over 1 million U.S. term infants, are expected to develop hypernatremia above 145 mEq/dL annually [1]. An estimated 5%, or 106,000, will develop moderate to severe hypernatremia (sodium ≥150 mEq/L) annually, 96% occurring in breastfed neonates [22]. Between 12 and 47% of those infants are expected to be moderately to severely disabled [5,59]. Given these estimates, approximately 13,000 to 50,000 U.S. term neonates are disabled by NHD annually. In addition, every year, approximately 3.7% or over 47,000 breastfed neonates are readmitted, the majority for feeding problems such as dehydration and hyperbilirubinemia, compared to 1.97% of formula-fed neonates [43,60]. The annual cost of excess readmission in breastfed neonates given an average cost of USD 14,300 per admission is over USD 678 million per year [61]. 

Hospital policies geared toward increasing exclusive breastfeeding rates before discharge may be contributing to cases of hypernatremia, particularly if no policy for hypernatremia screening or prevention exists. Given the serious nature of hypernatremia, it is important to take seriously any parental concerns regarding poor infant feeding or insufficient colostrum/milk production by providing a full clinical and laboratory assessment of infant feeding including evaluation of breast milk supply, breastfeeding efficacy, percent weight loss, and signs of infant satisfaction and hydration before assuming the infant is being adequately fed (see Table 4). Not all healthy, term infants can be adequately fed with colostrum alone and not all mothers produce enough colostrum. Exclusively breastfed neonates at the 95th percentile of weight loss can lose greater than 7–8% within 24 h [42]. A hospital policy that weighs infants every 12 h after birth can capture infants at the highest percentile of weight loss, who are at greatest risk for hypernatremia.

Sodium screening of infants with percent weight loss of 5% or greater, especially in neonates showing signs of inadequate feeding such as persistent crying and prolonged nursing, can help prevent hazardous levels of hypernatremia from developing. Greater than 5% weight loss during the birth hospitalization has been identified as an independent risk factor for neonatal jaundice readmission, a condition closely related to NHD [62]. Another study showed that increased breastfeeding assessment, infant weighing, and supplementation per a written hospital policy for any weight loss of ≥ 5% in 24 h reduced NHD cases by 37.5% [63]. A basic metabolic panel (BMP). and a serum or transcutaneous bilirubin (TcB) level for confirmed cases of hypernatremia and/or immediately before discharge, typically coinciding with the period of maximum weight loss, can also help identify breastfed and mix fed infants who are developing hypernatremia, hypoglycemia and hyperbilirubinemia and provide objective criteria for closer follow-up and recommendations for supplementation upon discharge. Sodium levels above 145 mEq/L should be clear indications for closer breastfeeding evaluation as well as supplementation, particularly if lactogenesis II has not occurred or if breast milk supply does not meet the full-term infant milk requirement of 5.5 oz/kg/day. Supplementation with banked donor milk (BDM) or formula to satisfaction can prevent the need for more invasive, costly, and potentially riskier correction with intravenous fluids. 

Since colostrum is typically produced in insufficient volumes to provide the full caloric and fluid requirement of a neonate (110 kcal/kg/day and 100 mL/kg/day, respectively) [19], small volumes of oral supplementation that mimic average colostrum volumes per day of life are unlikely to correct hypernatremia and other suboptimal feeding complications [44]. Therefore, it is recommended that for confirmed cases of hypernatremia between 145 and 150 mEq/L, alert infants should be offered supplemental BDM or formula until signs of infant distress resolve. Breastfeeding assessment can occur after the infant has been stabilized. In this situation, prevention of further weight loss and correction of hypernatremia should always be prioritized over minimizing exposure to supplemental formula. Banked donor breast milk should be used for premature or medically fragile neonates, and if available, can be used in hypernatremic term neonates where parents have a strong preference to avoid formula. Infants who are too weak due to dehydration and caloric deprivation are often ineffective at removing milk, which can threaten the breast milk supply. Supplemental milk may provide them with additional energy to breastfeed more effectively. Mothers whose infants are unable to remove milk effectively can also be assisted with instruction on manual expression of colostrum and double-electric breast pumping.

There may be concern that supplementation will compromise future breastfeeding. Uncontrolled formula supplementation without proper lactation management may lead to a reduction in breast milk removal and breast stimulation and subsequent secondary lactation failure [64]. In-hospital supplementation has been associated with earlier weaning of breastfeeding in multiple observational studies [65,66,67]. However, observational studies showing correlations between in-hospital supplementation and lower breastfeeding rates are retrospective, have non-standardized supplemental volumes and breastfeeding management algorithms and are unable to control for confounding variables such as biological factors that predict impaired lactation. They are unable to distinguish the couplets for whom supplementation is a sign of true low milk supply and inadequate feeding, which can predispose them to earlier breastfeeding cessation [68]. In other words, it is unclear from such data whether supplementation causes less robust milk supply or if less robust milk supply causes supplementation. The direction of causality between supplementation and earlier weaning can only be clarified by randomized controlled trials. 

The effects of judicious supplementation with 10 mL formula or banked human milk only after on-demand breastfeeding were tested in a total of five randomized controlled trials for weight loss of ≥5% or ≥75th percentile weight loss on the NEWT and all five found no reduction in any and exclusive breastfeeding rates at 3 months [69,70,71,72,73]. One study found no readmissions occurred in supplemented newborns and that supplementation had no effect on the infant gut microbiome [71]. Another found no effect on 6 month any and exclusive breastfeeding rates and a minimal decline in 1 year breastfeeding rates, which was confounded by shorter intended durations of breastfeeding that occurred by chance in the intervention group [72]. The data suggest that the observed association of in-hospital formula supplementation and earlier cessation of breastfeeding may either be related to uncontrolled formula supplementation without optimal breastfeeding management or true impaired lactation potential found in mothers who tend to supplement. Therefore, any in-hospital supplementation policy should protect the milk supply by ensuring adequate time feeding at the breast on-demand (8–12 times a day, 20–30 min/session) and optimal breast milk removal while providing supplemental volumes that can correct metabolic abnormalities. 

### 3.7. Management

Correction of NHD requires slow correction of sodium levels at about 0.5 mEq/L per hour, or no more than 12 mEqs per 24 h. A retrospective study of infants with NHD found that more rapid correction of hypernatremia of greater than 0.5 mEq/L/hour along with sodium concentration greater than 160 mEq/L were independent risk factors for convulsions or death [12]. As in the case discussed, it is possible for hypernatremic and hypoxic-ischemic brain injury to cause hypothalamic-pituitary infarction and subsequent SIADH, which makes gradual sodium correction even more difficult. Consultation with pediatric nephrology is recommended for cases of NHD. Detailed descriptions of hypernatremia correction strategies can be found in a review published by Durrani, et al. [76]. Correction of concurrent hyperbilirubinemia and hypoglycemia may also be required with phototherapy, exchange transfusion, and/or oral or IV dextrose as appropriate. 

Infants with sodium levels of 150 mEq/L and above should be offered a brain MRI during admission, given that 45.2% are positive for lesions consistent with injury at this level [11,77]. They should also be offered developmental follow-up evaluation given the known increased risk of developmental problems associated with sodium levels ≥ 150 mEq/L [56,57,58,59]. An EEG should be obtained for any seizure-like activity or signs of encephalopathy. Abnormal findings on MRI and/or EEG can help guide developmental follow-up, early developmental interventions, and anti-epileptic therapy (see Table 5).

It is also important to provide support for the family, who will likely experience trauma from unwittingly causing starvation-related complications while trying to comply with exclusive breastfeeding recommendations. Many will be grappling with the possibility of long-term disability of their previously healthy newborn. Some parents may reject breastfeeding due to this experience or choose combination feeding, and it is particularly important to respect parent autonomy by offering nonjudgmental support of their infant feeding choice. Lactating parents who wish to continue breastfeeding will require full evaluation of the milk supply, breast anatomy, and breastfeeding efficacy once the infant is stabilized. Twenty-four-hour measurement of milk supply through pumping and weighted feeds may be helpful in diagnosing problems with milk supply and milk transfer [36]. They will likely need assistance with pumping breast milk during their infant’s hospital stay, which should occur at least every 3 h. A five-hour period of sleep may be important to help the birthing parent recover and protect postnatal mental health, which can be accommodated with extra pumping sessions upon waking. Counseling on long-term combination feeding may be necessary in cases of persistent low milk supply.

## 4. Conclusions

Hypernatremic dehydration is one of the most costly and tragic complications that occur in previously healthy neonates. Newer data on hypernatremia now show that it is much more common than previously believed. The data also suggest that the long-accepted “normal” percent weight loss of 10% may overestimate a term neonate’s tolerance for weight loss. NHD is also one of the most preventable complications with some of the lowest cost interventions, namely USD 4 of formula or USD 60 of banked donor milk per day per affected infant. Even without higher cost testing such as for sodium levels and BMP, simple monitoring for weight loss, parent education on signs of persistent hunger and dehydration, and supplementation when signs of suboptimal feeding and ≥5% weight loss occur could prevent 98% of NHD cases. 

Clinical criteria for hypernatremia screening, prevention, and treatment need greater attention in hospital breastfeeding management algorithms. A comprehensive policy to prevent and diagnose NHD includes patient and health professional re-education on the signs and symptoms of suboptimal infant feeding, more frequent weight checks, laboratory screening when problem signs occur, pre-discharge laboratory screening, and discharge instructions on supplementation if an infant is at risk. Since more than a third of breastfed neonates will have hypernatremia before 72 h and more than 1 in 5 healthy mothers will have delayed lactogenesis II, such a comprehensive prevention program has the potential to save tens of thousands of infants from preventable feeding complications and readmissions. In addition, hypernatremia prevention has the potential to save hundreds of millions of dollars a year in health care costs for readmissions.

## Figures and Tables

**Table 1 children-09-01379-t001:** Mortality rates associated with severity of hypernatremia [8].

Sodium Concentration (mEq/L)	Mortality
150–160	3.6%
161–170	17.3%
171–189	66.6%

**Table 2 children-09-01379-t002:** Risk factors for neonatal hypernatremic dehydration and delayed lactogenesis II [16,25,26,27,28,29,30,31].

Neonatal Hypernatremic Dehydration	Delayed Lactogenesis II
Primiparity	Primiparity
Cesarean delivery	Cesarean delivery
Breast anomalies (surgery, no growth)	Flat or inverted nipples
Excessive pre-pregnancy weight	Maternal BMI > 27 kg/m^2^
Delayed first breastfeeding	Advanced maternal age ≥ 30 years
No prior breastfeeding experience	Hypertension
Ineffective breastfeeding latch/transfer	Endocrine problems: GDM, hypothyroid
Insufficient or absent colostrum Inadequate mature milk supply	Postpartum hemorrhage (Sheehan syndrome)
Delayed or failed lactogenesis II	Complicated, prolonged staged II labor

**Table 3 children-09-01379-t003:** Clinical presentation of hypernatremia [25,44,46,47,48].

Early Signs	Late Signs
Unsatisfied, frequent, prolonged nursing (>30–45 min/feed, <q2hrs)	Sleeping > 4 h w/o feeding
High-pitched, inconsolable crying	Sunken fontanelles
Body weight loss >5–7%	Lethargy
No wet diapers for 6 h (unreliable)	Kernicterus—opisthotonos, retrocollis
Pink or red dust in diapers (urate crystals)	Blank staring—encephalopathic facies
Short, ineffective, infrequent nursing	Hyperthermia
(<10 min, >q3hrs)	Seizures
Difficulty waking during feed	Hypothermia
Jaundice	Apnea, cyanosis
Poor skin turgor, dry lips	Bradycardia, cardiac arrest

**Table 4 children-09-01379-t004:** Prevention strategies.

Prevention Strategies	Recommendations
Weight loss monitoring	Weight check every 12 h after birth weight. Identify infants with weight loss >75th percentile on Newborn Weight Loss Tool and those with ≥5% weight loss for closer clinical and laboratory evaluation.
Sodium level screening	≥5% with risk factors and/or signs of suboptimal feeding.
Basic metabolic panel	Check BMP for sodium levels >145 mEq/L and for clinically significant hyperbilirubinemia (AAP) [74] or hypoglycemia (PES) [75].
Rescue supplementation	For alert, clinically dehydrated, distressed infants, offer supplementation after nursing with BDM or formula to satisfaction, if lactogenesis II has not occurred, while awaiting confirmation. Sub-threshold supplementation in response to early signs of hypernatremia can prevent the need for intravenous correction of hypernatremia, as well as phototherapy and/or exchange transfusion.
Pre-discharge screening	Weight check, BMP, and bilirubin for breastfed neonates.
Parent education	Assessment and counseling on risk factors for delayed lactogenesis II, low milk supply, and other breastfeeding challenges. Effective breastfeeding technique, signs of dehydration, hypernatremia, and suboptimal feeding; instruction on supplementation when signs occur followed by urgent pediatrician/LC visit.
Next day, frequent clinic follow-up	Next day and frequent visits until lactogenesis II and weight begins to increase. Evaluate for onset of lactogenesis II, breastfeeding frequency, duration, efficacy, infant behavior, signs of suboptimal feeding jaundice, dehydration, hypoglycemia, percent weight loss. LC evaluation of breastfeeding efficacy with weighted feed. Check BMP and bilirubin (TcB or TsB) for any signs of suboptimal feeding with ≥5% weight loss. Plan for supplementation for clinical or laboratory signs of suboptimal feeding present if lactogenesis II has not occurred or breast milk supply is inadequate to maintain weight. Counseling on protecting milk supply with adequate time breastfeeding, pumping for additional milk removal as needed. Hospital referral for hypernatremic dehydration ≥ 146 mEq/L, hyperbilirubinemia, hypoglycemia.

**Table 5 children-09-01379-t005:** Acute management of hypernatremia. Adapted with permission from Durrani et al. [76]. Copyright 2022, Durrani, N.U.R.; Imam, A.A.; Soni, N.

Hypernatremia Management
Emergent stabilization—20 mL/kg bolus until vitals stable, improved perfusion. Treatment goal—prevent cerebral edema by correcting sodium no faster than 0.5 mEq per hour or 12 mEq per day. Oral correction possible if sodium < 150 mEq/L with supplementation to satisfaction approaching full milk requirement of 5.5 oz/kg/day. Maintenance fluids—dextrose + 0.45% saline 100 mL/kg/day + 20 mL/kg/day if under radiant warmer. Glucose infusion rate of 5 mg/kg/min. Calculate free water deficit to desired goal of 10–12 mEq drop in sodium level per day (free water deficit = half of total volume of 0.45% saline solution per day). Sodium correction—every 2–4 h sodium checks until sodium is below 150 mEq/L; simultaneous 0.9% and 0.45% saline infusions. ○ If correction is too fast—increase 0.9% NS infusion and decrease 0.45% infusion by 20 mL/kg. ○ If correction is too slow—decrease 0.9% infusion and increase 0.45% infusion by 20 mL/kg. Extreme hypernatremia (>180 mEq/L) requires gradual correction with hypertonic saline; recommend nephrology consultation [76]. Treat comorbid hyperbilirubinemia per AAP hyperbilirubinemia guidelines and hypoglycemia per PES guidelines. NPO if lethargic; otherwise supplement infant as tolerated to infant satisfaction. Prioritize BDM for preterm or otherwise medically fragile infants. Obtain brain MRI for sodium ≥ 150 mEq/L, severe hyperbilirubinemia, symptomatic hypoglycemia, cardiovascular instability, need for cardiopulmonary resuscitation. (Case series of 96 hypernatremic neonate show a positivity rate of 45.2%). EEG for seizure-like activity, lethargy, blank staring, encephalopathy. Developmental/neurology follow-up for abnormal MRI, EEG or sodium ≥ 150 mEq/L or clinically significant hyperbilirubinemia or symptomatic hypoglycemia.

## Data Availability

Not applicable.

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
