# Peer review of "Fatal Hypernatremic Dehydration in a Term Exclusively Breastfed Newborn"

_children, 2022, doi:10.3390/children9091379_

Round 1

Reviewer 1 Report

The authors describe a case report of a full-term newborn with hypernatremic dehydration, followed by a narrative review of hypernatremic dehydration. The article is well written and makes valuable points. Hypernatremic dehydration in exclusively breastfed infants can be under-recognized and overlooked in certain situations with overzealous breastfeeding attempts. 

-The case presented here is clearly an unfortunate situation where an exclusively breastfed infant with risk factors of poor breastfeeding success ended up with the ultimate negative outcome or mortality. However, the evidence presented in this case can not conclusively say that exclusive breastfeeding was the only reason why this infant went into cardiac arrest. As the authors briefly mention, there may be conflicting explanations of what happened, including home/ sleep situation, underlying metabolic illness, etc. Without looking at this in detail, the strongest argument in this case, would be one of association, not causation. I also have some concerns about the language used in this manuscript - the tone of the article comes across strong against breastfeeding and may be better if toned down a little. 

-Case report description could be simplified with simply narrating facts and cut down interpretations, that may be more appropriate for discussion.  .

-Lines 56-58: "Delayed lactogenesis II" may need more introduction for some readers; kindly suggest that you introduce the concept briefly first. This is probably better suited for discussion than introduction / case details.

-Suggest remove "slow to pink" - Apgar scores of 8 & 9 are within normal limits for this infant. 

- The discussion, conclusion and recommendations are well written in good detail. Please review the discussion section to clarify causation vs association linking hypernatremia to breastfeeding in this infant. 

Author Response

Thank you so much for your very important and helpful feedback. I have added details to help illustrate how little the case infant received and how long he breastfed for. On the second day of life, there was no expressible colostrum. We have seen in our clinical practice that it is common for mothers to have lower to no colostrum production on the second day, which can result in a newborn receiving almost no fluid or calories for several days until they ultimately become too lethargic from dehydration and hypoglycemia if lactogenesis II does not occur in time. He also breastfed continuously for greater than 50 hours before he eventually became too lethargic to feed. I have added details in the manuscript to help illustrate the level of hunger he experienced. On a personal note, the lead author of this paper had a child who became unresponsive/lethargic at 15% weight loss with a sodium of 157 mEq/L. The literature also describes other infants become lethargic and apneic at similar weight loss percent due to hypoglycemia, which could have been a contributing factor (discussion now added to the manuscript). We have also reviewed a medicolegal case where the infant became lethargic and apneic with similar weight loss. The infant was weighed at 9.6% weight loss about 12 hours before becoming apneic with a glucose of 11 mg/dL with a moderate sodium level of 150 mEq/L. This infant also received minimal colostrum 1-4 mL and had documented fussiness and problems with feeding. This infant had no ketones on the urinalysis suggesting little to no energy substrate for cerebral brain function. So while the degree of hyperosmolarity and hypotension from moderate hypernatremia may not necessarily be severe enough to cause cardiac arrest on its own, the concurrent caloric deprivation and hypoglycemia that can occur from poor feeding due to delayed lactogenesis II may be sufficient to cause central apnea. If apnea is prolonged and undetected, it may be sufficient to cause brain stem injury as in the case infant discussed, and death from central apnea and cardiac arrest. While many cases of fatal hypernatremia occur past the first week of life, those infants likely received some measure of mature breast milk but too low to maintain weight and hydration. Those infants likely experienced more gradual declines in weight and increased in sodium levels to reach such extreme values. Infants who develop hypernatremia from delayed lactogenesis II are receiving much less milk per feed, approximately 0-5 mL, which provides only 3 kcal of a 300 kcal/day requirement (3 kg newborn). An infant experiencing such extreme caloric and fluid deprivation cannot be expected to survive more than a few days before developing apnea and cardiovascular collapse.

I have also described delayed lactogenesis II with a calculation of the expected caloric yield of exclusive colostrum feeding and why infants lose weight, dehydrate, and have rising sodium levels and declining glucose levels until lactogenesis II occurs.

I have also tried to tone down the language regarding breastfeeding. If you notice any specific sentences on the second round, I am happy to revise those sentences.

I have removed "slow to pink."

I have attached the revised document. Please note the detailed cost benefit analysis of a NHD prevention program.

Reviewer 2 Report

Overview:

Castillo-Hegyi et al. describe very carefully a complex and novel case of sudden newborn infant death. Throughout their article, we found no explanation for this case of lethal hypernatremia other than inadequate human milk intake. However, most (if not all) reported cases of neonates who were fed solely breast milk, developed severe dehydration and hypernatremia, and died, refer to jaundiced infants, with severe hypernatremia (>160mEq/L), admitted beyond day 8 of life, and with a weight loss beyond 15%. Since this study focus on a patient that did not meet the abovementioned inclusion criteria, which enable physicians to diagnose breastfeeding-associated fatal hypernatremic dehydration, the authors ought to clarify their reasons to discard other potential diagnoses and to consider inadequate mother’s milk intake as the main cause for the tragical end of this baby.

Particularly:

Line 274: “risk higher risk” to be corrected

Line 433: “Almost all cases of NHD are preventable with improved patient and health care professional training and with significant reforms in hospital policies”. We will thank enough evidence to support this statement.

Lines 439-440: “Infants experiencing weight loss of greater than 5%, particularly those with signs of inadequate feeding should be screened for hypernatremia, hypoglycemia, and hyperbilirubinemia.” Ardent attempts at maintaining breastfed neonates free from hypernatremia may cause more harm than good. Whilst the authors’ proposal would involve a huge number of neonates, a satisfactory cost-benefit analysis is needed in order to cope with such a new policy.

Author Response

"Castillo-Hegyi et al. describe very carefully a complex and novel case of sudden newborn infant death. Throughout their article, we found no explanation for this case of lethal hypernatremia other than inadequate human milk intake. However, most (if not all) reported cases of neonates who were fed solely breast milk, developed severe dehydration and hypernatremia, and died, refer to jaundiced infants, with severe hypernatremia (>160mEq/L), admitted beyond day 8 of life, and with a weight loss beyond 15%. Since this study focus on a patient that did not meet the abovementioned inclusion criteria, which enable physicians to diagnose breastfeeding-associated fatal hypernatremic dehydration, the authors ought to clarify their reasons to discard other potential diagnoses and to consider inadequate mother’s milk intake as the main cause for the tragical end of this baby."

I have added more detailed information regarding the infant's weight loss. Since the last weight was obtained more than 24 hours before being found unresponsive, I projected his weight loss to be around 14% if it followed the same linear pattern of weight loss as the preceding period of time since the mother did not experience lactogenesis II until the day after admission. He did not have a weight check before receiving an unknown amount of fluid from EMS and the ER. His first labs were done 5-6 hours after presentation to the ER after he had already received IV fluids and dextrose 10% infusion. So I posed the possibility that the sodium was higher than 155 mEq/L at the time of arrest and that there could have been concurrent hypoglycemia similar to other infants presenting between 2-5 days after birth, unresponsive from poor breastfeeding attempts with glucose levels below 36 mg/dL. These hypoglycemic infants develop extensive injury on brain MRI as well and had weight loss percent between 4 and 16%. The infant in the case sustained brain stem injury, which would cause central apnea. If he had developed this injury before becoming unresponsive and apneic, the amount of time his apnea was left undetected (30-45 minutes) could have been enough to cause cardiac arrest. While the level of hyperosmolarity and hypotension from moderate hypernatremic dehydration may not be enough to cause PEA and asystole (although we don't have data to know), cases we have reviewed and have seen in clinical practice, infants at 15% with even moderate hypernatremia and symptomatic hypoglycemia can become lethargic and apneic. On a personal note, the lead author's first born was also lethargic/unresponsive at 15% weight loss with a sodium of 157 mg/dL. Another infant whose medico-legal case we have reviewed was lethargic and apneic with a sodium of 150 with a glucose of 11 mg/dL. For this infant, the last weight check 12 hours before becoming severely symptomatic from hypoglycemia was 9.7%. This infant also have documented poor feeding episode and minimal intake 1-4 mL of colostrum per feeding. While other cases of fatal hypernatremia occur after the first week with higher percent weight loss, those infants likely had mothers who experienced lactogenesis II but had breast milk supplies that were insufficient to maintain weight and hydration. That would allow those infants to survive > 1 week with more gradual weight loss and slower rise in serum sodium. An infant receiving only scant colostrum of 1-5 mL per feed (which provides a maximum of 3 kcal of the 300 kcal/day requirement) is unlikely to survive more than 3-4 days of such severe caloric deprivation. 

With regard to jaundice, a systematic review shows only 45% of hypernatremic neonates are jaundiced.[1] In our clinical experience, non-Hispanic white infants are lower risk than Asian infants. So not all hypernatremic infants are jaundiced and clinicians should not include this possibility on the basis of an infant having no jaundice.

Particularly:

Line 274: “risk higher risk” to be corrected

I have corrected this.

Line 433: “Almost all cases of NHD are preventable with improved patient and health care professional training and with significant reforms in hospital policies”. We will thank enough evidence to support this statement.

I have provided a detailed cost-benefit analysis.

Lines 439-440: “Infants experiencing weight loss of greater than 5%, particularly those with signs of inadequate feeding should be screened for hypernatremia, hypoglycemia, and hyperbilirubinemia.” Ardent attempts at maintaining breastfed neonates free from hypernatremia may cause more harm than good. Whilst the authors’ proposal would involve a huge number of neonates, a satisfactory cost-benefit analysis is needed in order to cope with such a new policy.

I have written a more detailed justification for a NHD prevention program and a cost-benefit analysis. I will add the spreadsheet I used to calculate this. While unnecessary supplementation without adequate nursing time and stimulation of breast milk production can compromise breastfeeding success, five RCTS of judicious supplementation with 10 mL formula or donor milk after every breastfeeding session for ≥ 5% weight loss or ≥ 75%ile weight loss on the NEWT has shown no impact on 3 and 6 months exclusive and any breastfeeding rates. No RCT shows a negative impact of judicious supplementation on breastfeeding rates. Furthermore, hypernatremia is a medical indication for supplementation because it is a brain- and life-threatening condition that all infants should be protected from. Hypernatremic infants are often in distress and experience hours to days of suffering from hunger and thirst. Hospital policies that are permissive about allowing such serious complications to occur in order to increase exclusive breastfeeding rates may themselves be doing more harm than good. The known complications of hypernatremia are long-term disability and death. No such risks are associated with medically indicated and judicious preventative oral supplementation. If the point of infant feeding policy is to protect infant health and future brain development, then the priority should be ensuring infants receive enough caloric and fluid that protect their brain and vital organs. If health policy is failing to protect infants from these serious outcomes in order to comply with a hospital metric (like PC-05) that has been shown to increase morbidity (jaundice and dehydration) and not improve sustained breastfeeding rates, then we are failing to achieve the ultimate goal of protecting the health and safety for all infants. If we are to accept the current state standard of care and its complications, then all parents need to be provided informed consent on the increased risks associated with this strategy of breastfeeding promotion.

Supplementation of breastfeeding before lactogenesis II is common in many cultures where breastfeeding is the predominant form of infant feeding (Las Dos in Hispanic/Latino culture, pre-lacteal feeding in many cultures before the influence of Western exclusive breastfeeding advocacy) because they have been intrinsically aware of the dangers of inadequate infant feeding and starvation. Nigerian health care workers gave pre-lacteal feeds to prevent jaundice, dehydration, and hypoglycemia, which are now among the most common causes of neonatal readmissions. Despite wide use of pre-lacteal feeds, national average breastfeeding durations were 1-2 years of age. 

The aim of this work is not only to raise awareness of the common and serious complication of hypernatremia but to raise questions about our current strategy of breastfeeding promotion and its unintended harms. While infants whose mothers have robust milk supply may be protected and may even benefit from the current system of in-hospital exclusive breastfeeding promotion, a significant portion of infants who are least fed by this same strategy are exposed to unacceptable risks that their parents are not informed about. We have a duty to do no harm, to act in our patient's best interest, and to respect the autonomy of the parent by informing them of all the risks and benefits of every infant feeding strategy and every strategy to promote breastfeeding, including supplemented breastfeeding in response to signs of persistent infant hunger. I hope our additional text illustrates the importance of fulfilling this duty.

1.Lavagno C, Camozzi P, Renzi S, et al. Breastfeeding-Associated Hypernatremia: A Systematic Review of the Literature. J Hum Lact. 2016;32(1):67-74. doi:10.1177/0890334415613079

Round 2

Reviewer 1 Report

Thank you for addressing the reviewers' comments. The authors did a good job pointing out the facts of this case and discussing the interpretations with reasoning.

I feel that the language used to refer to hypernatremic dehydration “causing” cardiac arrest in a few places is not substantiated by the evidence in this case. I kindly recommend the authors review the manuscript in detail to identify such language that we may have missed. I still believe that this case highlights the need for safeguards during breastfeeding and this report is worth publishing. Given that this report will be hotly contested by breastfeeding advocates, it is important to avoid strong causative language and stick with available factual evidence.

A few minor comments:

1.       Abstract: Line 14: There is no proof that the infant developed hypernatremic dehydration over 63hrs of life. However, there is ample evidence in this case that the infant had problems with breastfeeding and had inadequate intake. Unfortunately, without checking sodium levels at the time, this becomes a hypothesis and not definitive proof of hyponatremic dehydration. May I suggest rephrasing this as “The infant “most likely” developed hypernatremic dehydration…”

2.       Abstract Line 15-16: “… cardiac arrest from hypernatremic dehydration…” – suggest rephrasing this as “…cardiac arrest, and found to have hypernatremic dehydration…”.

3.       Latch score – please add a qualifier to interpret latch scores for the readers – what is considered normal, what is poor, acceptable, not acceptable, etc.

4.       The cost-benefit analysis is very detailed. I am not convinced if this much detail is warranted. Also, the sources for the costs mentioned in each line item are not clear. Suggest the authors clarify the sources if they intend to include this detailed cost-benefit analysis in the final manuscript.

5.       It is unfortunate to hear that one of the authors had a negative personal experience with hypernatremic dehydration. It may be wise to include this in the conflict of interest statement at the end of the manuscript. 

Author Response

Comments and Suggestions for Authors

Thank you for addressing the reviewers' comments. The authors did a good job pointing out the facts of this case and discussing the interpretations with reasoning.

I feel that the language used to refer to hypernatremic dehydration “causing” cardiac arrest in a few places is not substantiated by the evidence in this case. I kindly recommend the authors review the manuscript in detail to identify such language that we may have missed. I still believe that this case highlights the need for safeguards during breastfeeding and this report is worth publishing. Given that this report will be hotly contested by breastfeeding advocates, it is important to avoid strong causative language and stick with available factual evidence.

Thank you for providing feedback on all sentences that may overstate causality. We agree with all the recommendations and have adjusted the language in the manuscript as recommended.

A few minor comments:

  1. Abstract: Line 14: There is no proof that the infant developed hypernatremic dehydration over 63hrs of life. However, there is ample evidence in this case that the infant had problems with breastfeeding and had inadequate intake. Unfortunately, without checking sodium levels at the time, this becomes a hypothesis and not definitive proof of hyponatremic dehydration. May I suggest rephrasing this as “The infant “most likely” developed hypernatremic dehydration…”
  2. Abstract Line 15-16: “… cardiac arrest from hypernatremic dehydration…” – suggest rephrasing this as “…cardiac arrest, and found to have hypernatremic dehydration…”.
  3. Latch score – please add a qualifier to interpret latch scores for the readers – what is considered normal, what is poor, acceptable, not acceptable, etc.

Thank you. We have written an explanation of the LATCH scoring system and interpretation of the numbers.

  1. The cost-benefit analysis is very detailed. I am not convinced if this much detail is warranted. Also, the sources for the costs mentioned in each line item are not clear. Suggest the authors clarify the sources if they intend to include this detailed cost-benefit analysis in the final manuscript.

We have provided a more detailed description of the numbers that were used to calculate the cost benefit analysis including a more succinct estimate of societal cost of child disability, which allowed me to remove a large section of the CBA. Weh ave also provided citations for each number used to calculate the costs and benefits. Please let me know if this version of the CBA section is more appropriate for publication.

  1. It is unfortunate to hear that one of the authors had a negative personal experience with hypernatremic dehydration. It may be wise to include this in the conflict of interest statement at the end of the manuscript.

From C.D.: I have written and highlighted a COI disclosure statement in the manuscript regarding this, but was not previously aware of any convention regarding investigators having to disclose a COI due to having a family member affected by the disease. I don’t recall ever reading any papers with such disclosure statements suggesting this as a conflict of interest. I would request the input of the journal’s editors regarding the appropriate course regarding this matter.

Reviewer 2 Report

Question 1

the authors ought to clarify their reasons to discard other potential diagnoses and to consider inadequate mother’s milk intake as the main cause for the tragical end of this baby

The authors support the inclusion of this baby in the group of fatal hypernatremia linked to insufficient breastfeeding based on personal notes, the text should specify this point, as well as their guess about sodium and glucose blood levels previous to resuscitation.

Question 2

Line 433: “Almost all cases of NHD are preventable with improved patient and health care professional training and with significant reforms in hospital policies”. We will thank enough evidence to support this statement.

I have provided a detailed cost-benefit analysis.

A cost-benefit analysis might support strategies to prevent fatal hypernatremia in the future, but the point here is that I kindly ask for references from the past to support this statement: “Almost all cases of NHD are preventable with improved ... In case that there is not a group that was able to improve their figures on fatal hypernatremia following new policy implementation, this sentence should be erased.

Question 3

Ardent attempts at maintaining breastfed neonates free from hypernatremia may cause more harm than good. Whilst the authors’ proposal would involve a huge number of neonates, a satisfactory cost-benefit analysis is needed in order to cope with such a new policy.

I have written a more detailed justification for a NHD prevention program and a cost-benefit analysis.

This is a carefully written cost-benefit analysis that does not take into account the potential effect of this proposal on breastfeeding cessation. Since breastfeeding cessation has detrimental effects on the mother and on the child in the short, medium and long term, this variables must be included in the analysis.

Author Response

Question 1

the authors ought to clarify their reasons to discard other potential diagnoses and to consider inadequate mother’s milk intake as the main cause for the tragical end of this baby

The authors support the inclusion of this baby in the group of fatal hypernatremia linked to insufficient breastfeeding based on personal notes, the text should specify this point, as well as their guess about sodium and glucose blood levels previous to resuscitation.

Thank you. I have included in the text our estimate of the infant’s expected sodium level (160-170 mEq/L) and glucose level (<20 mg/dL) similar to other infants developing lethargy and apnea related to hypernatremia and hypoglycemia, respectively, at his expected percent weight loss (14%).

Question 2

Line 433: “Almost all cases of NHD are preventable with improved patient and health care professional training and with significant reforms in hospital policies”. We will thank enough evidence to support this statement.

I have provided a detailed cost-benefit analysis.

A cost-benefit analysis might support strategies to prevent fatal hypernatremia in the future, but the point here is that I kindly ask for references from the past to support this statement: “Almost all cases of NHD are preventable with improved ... In case that there is not a group that was able to improve their figures on fatal hypernatremia following new policy implementation, this sentence should be erased.

Thank you. We agree. We had removed that statement in second version of the manuscript after the first round of reviews, since we don’t know how many cases of NHD can be prevented without studies. Furthermore, real life clinical practice does not actually provide such complete protection from bad outcomes. We have opted for less strongly worded estimates of the potential impact of a NHD prevention program.

Question 3

Ardent attempts at maintaining breastfed neonates free from hypernatremia may cause more harm than good. Whilst the authors’ proposal would involve a huge number of neonates, a satisfactory cost-benefit analysis is needed in order to cope with such a new policy.

I have written a more detailed justification for a NHD prevention program and a cost-benefit analysis.

This is a carefully written cost-benefit analysis that does not take into account the potential effect of this proposal on breastfeeding cessation. Since breastfeeding cessation has detrimental effects on the mother and on the child in the short, medium and long term, this variables must be included in the analysis.

I have written two paragraphs in the conclusion to justify our analysis and our hypothesis that a program that provides only judicious and controlled supplementation after adequate time nursing has been shown to have no statistically significant impact on 3 and 6 month breastfeeding rates and indeterminant effects on 1 year breastfeeding rates. Since all 5 of the currently existing randomized controlled trials on the effects of such practice with either formula or banked donor milk and no RCTs show a negative effect, I am uncertain how we can calculate the cost of lower rates of breastfeeding without a percent reduction figure. All other studies are observational and measure the effects of non-standardized, uncontrolled supplementation of breastfeeding, with unclear management of breastfeeding (unclear breast milk removal/efficacy), unknown reasons for supplementation, and unclear measurements of confounding variables that affect lactation, like insufficient glandular tissue signs, GDM, high BMI, etc. The only way to control for those variables is through a standardized supplementation protocol for medically indicated conditions like hypernatremia and randomization. If we are missing data that can help guide us to provide a more complete CBA, we would be glad to be directed to such data so that we can provide a more complete analysis.